# Discriminating North American Swine Influenza Viruses with a Portable, One-Step, Triplex Real-Time RT-PCR Assay, and Portable Sequencing

**DOI:** 10.3390/v16101557

**Published:** 2024-09-30

**Authors:** Marie K. Kirby, Bo Shu, Matthew W. Keller, Malania M. Wilson, Benjamin L. Rambo-Martin, Yunho Jang, Jimma Liddell, Eduardo Salinas Duron, Jacqueline M. Nolting, Andrew S. Bowman, C. Todd Davis, David E. Wentworth, John R. Barnes

**Affiliations:** 1Virology, Surveillance and Diagnostic Branch, Influenza Division, Centers for Disease Control and Prevention, Atlanta, GA 30329, USA; pbi0@cdc.gov (M.K.K.); brs9@cdc.gov (B.S.); szt0@cdc.gov (E.S.D.);; 2Chippewa Government Solutions (CGS), Sault Tribe Incorporated (STI), Cherokee Nation Operational Solutions (CNOS), Sault Sainte Marie, MI 49783, USA; 3Department of Veterinary Preventive Medicine, The Ohio State University, Columbus, OH 43210, USA

**Keywords:** swine influenza virus, triplex real-time RT-PCR, portable sequencing

## Abstract

Swine harbors a genetically diverse population of swine influenza A viruses (IAV-S), with demonstrated potential to transmit to the human population, causing outbreaks and pandemics. Here, we describe the development of a one-step, triplex real-time reverse transcription-polymerase chain reaction (rRT-PCR) assay that detects and distinguishes the majority of the antigenically distinct influenza A virus hemagglutinin (HA) clades currently circulating in North American swine, including the IAV-S H1 1A.1 (α), 1A.2 (β), 1A.3 (γ), 1B.2.2 (δ1) and 1B.2.1 (δ2) clades, and the IAV-S H3 2010.1 clade. We performed an in-field test at an exhibition swine show using in-field viral concentration and RNA extraction methodologies and a portable real-time PCR instrument, and rapidly identified three distinct IAV-S clades circulating within the N.A. swine population. Portable sequencing is used to further confirm the results of the in-field test of the swine triplex assay. The IAV-S triplex rRT-PCR assay can be easily transported and used in-field to characterize circulating IAV-S clades in North America, allowing for surveillance and early detection of North American IAV-S with human outbreak and pandemic potential.

## 1. Introduction

Influenza A virus (IAV) is a respiratory pathogen that results in millions of infections globally each year, and pandemics from IAVs continue to be a major threat to public health. IAV in swine are of interest from a public health perspective, as swine are susceptible to infection from both human-origin and avian-origin IAVs [1,2]. Coinfections of a host with multiple IAVs can lead to reassortment events, in which two or more different viral strains infect the same host cell and exchange gene segments, creating genetically and potentially antigenically distinct viral strains. Due to their susceptibility to influenza viruses from multiple species, swine are reservoirs for genetically distinct viruses to which certain human populations may be immunologically naïve [3]. IAVs can bidirectionally transmit between humans and swine, adding to the complexity of swine IAV (IAV-S) evolution, as human IAVs can be sequestered within swine populations and re-emerge into a human population that is vulnerable to infection. The 2009 influenza pandemic and the human infections of influenza A(H3N2) variant (v) [(H3N2)v] IAVs that originate in swine, have demonstrated that reassortant IAV-S can arise with the ability to cause outbreaks of illness and pandemics in human populations [3,4,5,6,7].

IAV-S A(H1) 1A clade virus, or so-called classical IAV-S (cIAV-S) A(H1N1), was first isolated from swine in 1930 in the United States (U.S.) and remained antigenically stable until the 1990s [8,9,10,11]. In the late 1990s, the cIAV-S A(H1N1) reassorted with a human A(H3N2) IAV and a North American (N.A.) avian IAV to create a triple reassortant A(H3N2) IAV-S, which spread rapidly throughout the N.A. swine population [2,12,13]. Shortly thereafter, triple reassortant A (H1N1) and A (H1N2) IAV-S were also observed circulating in N.A. swine populations [10,12,14]. These reassortant IAV-S all contained the same internal set of genes, termed the triple-reassortant internal gene (TRIG) cassette, with the polymerase basic 2 (PB2) and polymerase acidic (PA) genes from N.A. avian IAV origin, nucleoprotein (NP), matrix (M) and nonstructural (NS) genes from cIAV-S origin, and polymerase basic 1 (PB1) gene from human IAV origin, which combined with distinct hemagglutinin (HA) and neuraminidase (NA) genes to create genetic diversity in the circulating IAV-S population [15]. In the mid-2000s, human A(H1N1) and A(H1N2) IAVs crossed into the N.A. swine population and created reassortant viruses with HA genes that were genetically and antigenically distinct from other circulating H1 IAV-S, and these were termed the 1B.2.2 (H1δ1) and 1B.2.1 (H1δ2) clades [11,16,17,18]. After causing the first influenza pandemic of 21st century, the 2009 A(H1N1) pandemic influenza [A(H1N1)pdm09] virus subsequently was transmitted back into swine and created a distinct gene constellation with the same TRIG cassette, but with the M gene from the A(H1N1)pdm09 virus, originally acquired from a Eurasian lineage IAV-S [3,19]. Reassortment events have continued between the A(H1N1)pdm09 virus and established IAV-S, creating increased genetic diversity in the IAVs circulating in N.A. swine [20,21,22]. Currently, at least 10 distinct HA genetic clades cocirculate in N.A. swine, including the IAV-S A(H1) 1A.1 (α), 1A.2 (β), 1A.3 (γ), 1B.2.2 (δ1), 1B.2.1 (δ2) clade, 1A.3.3.2 (A(H1N1)pdm09) clade, and the IAV-S H3 1990.4, 2010.1, and 2010.2 clades [6,11,20,23].

From a public health perspective, increased genetic diversity in circulating IAV-S creates a threat to humans, as new reassortant and new variant viruses can increase the likelihood of an outbreak or pandemic. Viruses that contain genes from older human strains against which segments of the human population have no immunological memory are of greatest concern. For example, in humans, the A(H1N1)pdm09 virus has been the only circulating A(H1N1) virus in a decade, and it has been the selected A(H1N1) component of the WHO-recommended influenza virus vaccine since 2010. As there is poor antigenic cross-reactivity between the A(H1N1)pdm09 virus and the 1B.2.2 (H1δ1) and 1B.2.1 (H1δ2) clades, this creates a substantial antigenic gap, especially in children that have no prior exposure to human A(H1N1) viruses that circulated before 2009, represented by the A/Brisbane/59/2007 A(H1N1) virus [22]. Additionally, in 2016, there was an outbreak among persons who attended agricultural fairs in the mid-west U.S. of an A(H3)v 2010.1 clade virus that contained an HA gene that was newly detected in IAV-S H3 viruses, likely introduced into N.A. swine from humans during a reverse zoonotic transmission in 2010 or 2011 [7,24]. Therefore, it is important to have a rapid detection method for IAV-S A(H1) 1B.2.2 (δ1) and A(H1) 1B.2.1 (δ2) clades and IAV-S A(H3) 2010.1 clade viruses during swine influenza outbreaks, as these are viruses that have a high likelihood for causing outbreaks if transmitted to humans.

Here, we describe a one-step, triplex real-time reverse transcription-polymerase chain reaction (rRT-PCR) assay that detects and distinguishes IAV-S A(H1) 1A.1 (α), 1A.2 (β), 1A.3 (γ), 1B.2.2 (δ1) and 1B.2.1 (δ2) clades, and IAV-S A(H3) 2010.1 clade viruses circulating in N.A swine [25]. The IAV-S triplex rRT-PCR (IAV-S triplex rRT-PCR) assay consists of three sets of primers and three dual-labeled hydrolysis probes each targeting different regions of the IAV-S HA gene, and this multiplex assay has high sensitivity and specificity to detect and distinguish these antigenically distinct HA clades. We adapted the instrumentation to mobilize this assay for in-field use directly at interfaces of high zoonotic transmission potential, such as agricultural fairs and swine exhibitions. We demonstrate the portability of this assay via an in-field test at an exhibition swine show. We used in-field viral concentration and RNA extraction methodologies and a portable real-time PCR instrument, the Quantabio “Q” qPCR instrument, and rapidly identified three distinct IAV-S clades circulating within the N.A. swine population. Therefore, the IAV-S triplex rRT-PCR assay is a portable resource that can rapidly and sensitively monitor and characterize circulating N.A. IAV-S to identify clades that pose a high risk for causing human outbreaks and potential pandemics.

## 2. Materials and Methods

### 2.1. Influenza Viruses and In-Lab RNA Extraction

Influenza viruses used for testing of the IAV-S triplex rRT-PCR assay were grown to high titer in either Madin-Darby Canine Kidney (MDCK) cells or embryonated chicken eggs (ECE) [26]. Tissue culture infectious dose-50% (TCID_50_/_mL_) or egg-infectious dose-50% (EID_50_/_mL_) were used to measure infectious virus titer in the cell supernatant or allantoic fluid, respectively. Viral RNA was extracted from 120 µL cellular supernatant or allantoic fluid using the Qiagen EZ1 DSP Virus Kit and the Qiagen EZ1 Advanced XL automated extractor. Total nucleic acid was eluted in 120 µL of Qiagen RNA elution buffer.

### 2.2. Swine Influenza Triplex Assay

#### 2.2.1. Swine Influenza Triplex Assay Primers and Probes

The cH1, sH1, and H3v_2010 primer and probes were selected from highly conserved regions of the IAV-S H1 HA gene or the HA gene of IAV-S H3 2010.1 clade viruses based on each available clade nucleotide sequence data from GenBank database of National Centers for Biological Information (NCBI), the Global Initiative on Sharing All Influenza Data (GISAID) and the Influenza Research Database (ISD) at the time of development of this assay. All available HA sequences for each clade were downloaded and aligned. Poor-quality sequences were removed. Primer and probe sequence specificity was also evaluated throughout NCBI, GISAID, and ISD databases. Primers and probes were designed to have annealing temperatures of approximately 60 °C and 70 °C, respectively, using PrimerExpress 3.0.1 software (Applied Biosystems, Foster City, CA, USA). Primer and probes were synthesized by the Biotechnology Core Facility at the CDC (Table 1).

The sH1 probe was labeled at the 5′-end with the reporter molecule 6-carboxyfluorescein (FAM) and quenched with Blackhole Quencher™ 1 (BHQ™1) internally at a modified “T” residue with a modified 3′-end (Spacer 6) to prevent probe extension by Taq polymerase. The H3v_2010 probe, a BHQplus^TM^ dual-labeled hydrolysis probe, was labeled at the 5′-end with hexachlorofluorescein (Hex), with the Blackhole Quencher™ 1 (BHQ™1) located at the 3′-end and employed C-5 propynyl-dC (pdC) for dC and C-5 propynyl-dU (pdU) for dT substitutions (Glen Research Corporation, Sterling, VA, USA). The cH1 probe was labeled at the 5′-end with Quasar 670 (Triplex, Cy5 channel) and quenched with a BHQ™2 internally at a modified “T” residue with the addition of a Spacer 3 (C3) modification at the 3′-end (Spacer 3, Catalog: 3-Sp3, Biosearch Technologies, Inc., Novato, CA, USA).

During the COVID-19 pandemic, our lab developed the FluSC2 Multiplex assay [27]. We have subsequently built the swine triplex leveraging some of the fluorophore and quencher combinations we used for the FluSC2 build and found an enhanced limit of detection for the swine triplex versus the initial build described in this manuscript. For this updated build, the sH1 probe is labeled at the 5′-end with the reporter molecule FAM, a nova quencher between base residues 9 and 10, and a BHQ™1 quencher at the 3′ end. The H3v_2010 probe is the same as described above, except the fluorophore is CIV-550 in place of HEX. The cH1 probe has a nova quencher between base residues 9 and 10, and a BHQ™1 quencher at the 3′ end.

#### 2.2.2. In-Lab rRT-PCR Reaction Conditions

Reaction conditions for rRT-PCR were based upon conditions for the FDA-cleared CDC Human Influenza Virus Real-Time RT-PCR Diagnostic Panel (CDC rRT-PCR Flu Panel), which allows for simultaneous testing with other CDC diagnostic assays [28,29]. Single-plex rRT-PCR reactions were performed using SuperScript™III Platinum^®^ One-Step quantitative RT-PCR (qRT-PCR) Kits (Cata: 11732088, Life Technologies, Carlsbad, CA, 92008, USA) as described previously [29], and multiplex rRT-PCR reactions were optimized and performed using TaqPath™ 1-Step Multiplex Master Mix (Cata: A28522, Life Technologies, Carlsbad, CA, 92008). Multiplex rRT-qPCR assays were performed in a final volume of 25 μL, including 6.25 μL of TaqPath™ 1-Step Multiplex Master mix (4×) and 5 μL of RNA. Primers were added at a final concentration of 800 nM and probes at 200 nM. rRT-PCR thermocycling conditions were as follows: 50 °C for 30 min, Taq activation at 95 °C for 2 min, and 45 cycles of 95 °C for 15 sec and 55 °C for 30 sec. rRT-PCR reactions were measured on the Applied Biosystems™ 7500 Fast Dx Real-Time PCR (AB 7500 Dx; Life Technologies, Carlsbad, CA, 92008, USA) instrument.

#### 2.2.3. Sensitivity and Specificity

The sensitivity of the IAV-S trRT-PCR assay was determined via testing on the AB 7500 Fast Dx Real-Time PCR instrument using three IAV-S representing three genetic clades, including A/Ohio/35/17(H1N2)v (H1 1B.2.1 clade), A/Ohio/27/2016(H3N2)v (H3 2010.1 clade), and A/Ohio/24/2017(H1N1)v (H1 1A.1.1 clade), in comparison to the universal influenza A (InfA) single-plex assay from the CDC Flu rRT-PCR Dx Panel for universal detection of M gene of all IAVs [28,29] (Table 2 and Appendix A). The sensitivity and specificity of the assay were further evaluated by testing the IAV-S triplex rRT-PCR assay against 1A.1.1, 1A.2, 1A.3.3.3, 1B.2.2, and 1B.2.1 clades of IAV-S H1; clades 1990.4, 2010.1, and 2010.2 of IAV-S H3 viruses; human A(H1N1), A(H1N1)pdm09, seasonal A(H3N2), HPAI A(H5N1), HPAI A(H5N6), Eurasian lineage A(H7N9), influenza B/Yamagata, and B/Victoria lineage viruses (Table 3 and Table 4).

#### 2.2.4. Adaptation for Portable Testing

Performance of the IAV-S triplex rRT-PCR assay on the “Q” qPCR instrument was evaluated via comparisons to performance of the IAV-S triplex rRT-PCR assay on the AB 7500 Dx instrument (Table 5). The IAV-S triplex rRT-PCR assay was performed on both instruments with RNA extracted from five IAV-S representing three viral clades and one human A(H1N1), including A/Ohio/35/17(H1N2)v (H1 1B.2.1 clade), two A(H3N2) viruses (H3 2010.1 clade) A/Ohio/27/2016 and A/Ohio/28/2016clade, A/Ohio/24/2017(H1N1)v (H1 1A.1.1 clade), A/Texas/14/2008 (H1 1A.2 clade), and A/Brisbane/59/2007 (human A(H1N1)). Data analysis for each target on the “Q” qPCR instrument was set as: Method = dynamic; Ignore Cycles Before = 3, Threshold level = 1.0, Exclusion = Extensive, and Fluorescence Cutoff Level = 5%.

To test if in-field RNA extraction methodology would impact performance of the IAV-S triplex rRT-PCR assay, RNA was extracted from IAV-S representing three viral clades (A/Michigan/383/2018—1B.2.1; A/Indiana/27/2018—H3v 2010.1; A/Ohio/9/2015—1A.3.3.3) using Akonni TruTip Rapid RNA extraction Kit (Akonni Biosystems, Frederick, MD, USA) according to the manufacturer’s instructions using a manual 12-channel p1000 pipette. RNA was eluted in 75 µL of Akonni elution buffer in strip tubes, and 1:10 serial dilutions were created for each virus (range: 10^−1^ to 10^−5^). The resulting RNA dilutions were tested on the “Q” qPCR instrument with the IAV-S triplex rRT-PCR assay (Appendix A).

### 2.3. In-Field Sample Collection, RNA Extraction, and rRT-PCR Detection

Nasal swabs were collected from 132 swine: 96 swabs on day 1 and 36 additional swabs on day 2. On day 1, swabs were initially collected from 96 swine exhibiting signs of ILI using polyester-tipped swabs, and the sampling swabs were placed immediately in viral transport media (BD Universal Viral Transport 3 mL Vial kit). To enrich the virus concentration prior to nucleic acid extraction, 1.5 mL of each sample was mixed with 100 µL Nanotrap red magnetic particles (Ceres Nano, Manassas, VA, USA) and incubated for 20 min in a 96-deep well block [30]. Nanotrap beads were then pelleted using a magnet, and the supernatant was discarded. Viral RNA was extracted directly from the Nanotrap beads using the Akonni TruTip Rapid RNA extraction Kit (Akonni Biosystems, Frederick, MD, USA) according to the manufacturer’s instructions using a manual 12-channel p1000 pipette. Extracted RNA for each sample was eluted in 75 µL of Akonni elution buffer in strip tubes. An additional 36 samples were subsequently collected from swine housed in close proximity to swine swabbed during the first sampling with a Ct value of <33 for InfA (InfA+). These additional 36 swine were sampled via sterile gauze nasal wipes submerged in 5 mL brain heart infusion (BHI) media [31,32]. Nanotrap viral enrichment and nucleic acid extraction were completed for these 36 additional samples as described above. rRT-PCR was run in-field using a portable “Q” qPCR instrument. Subtype/Clade specific results from the IAV-S triplex rRT-PCR assay were compared to in-field sequencing data.

### 2.4. Portable Sequencing and Sequence Analysis

Sequencing was performed in-field by our mobile influenza analysis (*Mia*) platform as described previously [33]. Briefly, the full influenza genome was amplified via fast multisegmented reverse transcription polymerase chain reaction (fMRT-PCR). Barcodes were added to the amplicons with further PCR cycling using primers containing the barcode sequences. Following barcoding, sequencing adapters were ligated to pooled amplicons using the SQK-LSK 108 library kit (Oxford Nanopore Technologies, Oxford, UK). The prepared library was sequenced on a MinION Mk 1B nanopore sequencer using flow cell FAK49169 equipped with R9.4.1 (FLO-MIN106) chemistry. Raw fast5 read files were base called and demultiplexed with Guppy v.2.3.7 using default parameters. Reads were mapped and assembled into influenza genomes using IRMA v.0.6.7 with a MinION configuration module. Plurality consensus sequences for each segment were used in analyses.

## 3. Results

### 3.1. Establishing the Swine Influenza Triplex rRT-PCR Assay

The IAV-S triplex rRT-PCR assay is a multiplex rRT-PCR assay developed for rapid detection of 1A.1, 1A.2, 1A.3, 1B.2.2, and 1B.2.1 clades of IAV-S A(H1) viruses and IAV-S A(H3) 2010.1 clade viruses circulating in the N.A. swine population, and it can be used during surveillance of influenza outbreaks within N.A. swine populations. The IAV-S triplex rRT-PCR assay consists of the following single-plex rRT-PCR assays: cH1, sH1, and H3v_2010, which are used by the Influenza Division at CDC for detection of swine-origin influenza virus variants in human specimens that originate in swine, including 1A.1, 1A.2, 1A.3, and 1B.2 clades of A(H1)v and A(H3)v 2010.1 clade viruses (Table 1, Appendix A). The assay was developed into a multiplex rRT-PCR design to allow for efficiency for portable, in-field testing.

The sH1 assay was designed to detect IAV-S A(H1) 1B.2.2 (δ1), 1B.2.1(δ2) lineage viruses, and 1B.2-like viruses, including human A(H1N1) viruses that circulated prior to 2009 (Appendix A). The H3v_2010 assay was designed to detect IAV-S A(H3) 2010.1 clade viruses (Appendix A). The cH1 assay was designed to detect IAV-S A(H1) 1A.1 (α), 1A.2 (β), and 1A.3 (γ) clades. In order to better discriminate against A(H1N1)pdm09 viruses, which inherited their HA gene from the IAV-S A(H1) 1A.3.3.3 (γ) clade, a nucleotide change from thymine to cytosine at the tenth nucleic acid from the 5′-end was introduced into the cH1 probe (Appendix A).

### 3.2. Sensitivity and Specificity of Swine Influenza Triplex rRT-PCR Assay

The performance of the IAV-S triplex rRT-PCR assay was evaluated via comparisons with the sensitivity of the universal influenza A (InfA) assay from the CDC Flu rRT-PCR Dx panel to detect 10-fold serial dilutions of relevant viruses, including A/Ohio/35/17(H1N2)v (1B.2.1 clade), A/Ohio/27/2016(H3N2)v (H3 2010.1 clade), and A/Ohio/24/2017(H1N1)v (1A.1.1 clade). The limit of detection of the sH1, H3v_2010, and cH1 components in the context of the IAV-S triplex assay was 10^3.9^, 10^4.1^, and 10^3.3^, respectively (Table 2 and Appendix A).

We assessed the performance and specificity of the IAV-S triplex rRT-PCR assay with a diverse population of high-titer IAV-S isolated from swine and humans, testing RNA extracted from thirteen N.A. A(H1) and A(H3) IAV-S, including 1A.1, 1A.2, 1A.3, 1B.2.2, and 1B.2.1 clades and A(H3) clades of 1990.4, 2010.1, and 2010.2 viruses, and one Eurasian IAV-S A(H1) virus (1A.1.4) (Table 3). As expected, all three N.A. A(H1) 1A.1 viruses, including two 1A.1.1, two H1 1A.2, and one A(H1) 1A.3.3.3 viruses, were positive for the cH1 target, and cross-reactivity was not observed for the H1-1B.2 viruses or the three clades of A(H3) viruses that were tested. Three A(H1)-1B.2 viruses, including two 1B.2.2 viruses and one 1B.2.1 virus, were positive for the sH1 target, and no cross-reactivity was observed when the H1-1A.1, H1-1A.2, H1-1A.3, and the three clades of A(H3) viruses were tested. A human pre-2009 A(H1N1) virus, A/Brisbane/59/2007 (1B.2), tested positive for the sH1 target only. Two A(H3) 2010.1 clade viruses were positive for the H3v_2010 target, while, as expected, all IAV-S H1 clade viruses, one H3 2010.2 clade virus and two H3 1990.4 clade viruses introduced into the swine population in the late 1990s were negative [34]. RNA extracted from an A(H1) 1A.1.4 clade virus from Thailand was negative for all three targets of the IAV-S triplex rRT-PCR assay. The Ct values of viruses positive for the IAV-S triplex rRT-PCR assay were comparable to the Ct value of the InfA assay (<3 Cts difference between the InfA assay and the positive target of the IAV-S triplex rRT-PCR) (Table 3).

To ensure exclusivity performance of the IAV-S triplex rRT-PCR assay, we tested the IAV-S triplex rRT-PCR assay on high titer A(H1N1)pdm09, seasonal A(H3N2), highly pathogenic avian influenza (HPAI) A(H5N1) and A(H5N6), Eurasian lineage A(H7N9), influenza B/Yamagata and B/Victoria lineage viruses (Table 4). The IAV-S triplex rRT-PCR assay showed no positive signal towards these viruses, whereas the IAVs were positive for the InfA rRT-PCR assay, and the influenza B viruses were positive for the influenza B (InfB) rRT-PCR assay from the CDC Flu rRT-PCR Dx panel.

### 3.3. Testing Instruments and Methodologies for Portability

The performance of the IAV-S triplex rRT-PCR assay on the “Q” qPCR instrument was evaluated via comparisons to the performance of the IAV-S triplex rRT-PCR assay on the AB 7500 Dx instrument (Table 5). We performed the IAV-S triplex rRT-PCR assay on RNA extracted from IAV-S representing three IAV-S clades and one human A(H1N1) IAV. Viruses tested included A/Ohio/35/17(H1N2)v (1B.2.1 clade), A/Ohio/27/2016 and A/Ohio/28/2016(H3N2)v (H3 2010.1 clade), A/Ohio/24/2017(H1N1)v (1A.1 clade), A/Texas/14/2008(H1N1)v (1A.2 clade), and A/Brisbane/59/2007 (human pre-2009 H1N1). We observed comparable data (≤1 Ct difference between instruments for each target) from both instruments with the IAV-S triplex rRT-PCR assay (Table 5).

To test if in-field RNA extraction methodology would impact the performance of the IAV-S triplex rRT-PCR assay, RNA was extracted from IAV-S representing three viral clades using Akonni TruTip Rapid RNA extraction Kit (Akonni Biosystems, Frederick, MD, USA) and subsequently tested on the portable “Q” qPCR instrument with the IAV-S triplex rRT-PCR assay. We observed successful extraction and detection using this portable methodology and instrumentation (Appendix A).

### 3.4. Performance of Swine Influenza Triplex rRT-PCR Assay In-Field

We tested the in-field performance of the IAV-S triplex rRT-PCR assay at a large agricultural fair, sampling a total of 132 swine, with 96 sampled on day 1 and an additional 36 sampled on day 2 (Appendix A). On day 1, we initially sampled 96 swine, including any swine that we observed to be exhibiting signs of influenza-like illness (ILI). The swabs were placed in a viral transport medium, and RNA extractions were performed on-site within 1–2 h after sampling was complete. Prior to RNA extraction, samples were preincubated with Nanotrap beads to concentrate viruses. Viral RNA was subsequently extracted directly from the Nanotrap beads using Akonni TruTip RNA extraction kits.

We initially tested 92 of these first 96 extractions in-field for influenza A positivity using the CDC InfA rRT-PCR single-plex assay and the portable “Q” qPCR instrument and identified 21 positive swine samples. The cutoff Ct value for rRT-PCR, including the IAV-S triplex assay, was set to be <33 for this field study, as higher Ct values would be less likely to have enough nucleic acid to be detected by subsequent in-field MinION sequencing [33]. On day 2, we sampled an additional 36 swine housed in close proximity in the barn to the swine with InfA positive RT-PCR results. We tested the 21 InfA positive RNA samples from the first round of extractions, three RNA samples from the first extractions that were not preliminarily tested with InfA, and the 36 new RNA extractions, for a total of 60 RNA samples, with the IAV-S triplex Assay. The fourth sample from day 1 was not included in day 2 testing due to a known user error.

From these 60 specimens, using the IAV-S triplex assay, we identified 18 swine were positive for the cH1 target and 6 swine were positive for the sH1 target (Table 6 and Appendix A). There were no IAV-S H3 2010.1 clade viruses detected by the IAV-S triplex rRT-PCR assay.

These 60 samples were subsequently sequenced in-field using the *Mia* platform developed by the Influenza Division/CDC and run on an Oxford Nanopore Technologies Minion portable sequencing flow cell, as described previously [33]. Sequencing confirmed the results from the IAV-S triplex rRT-PCR assay, demonstrating that the main viral populations circulating in the exhibition swine consisted of IAV-S A(H1)-1B.2.1, A(H1)-1A.3.3.3, and A(H1N1)pdm09 (1A.3.3.2) clades, with no IAV-S H3 2010.1 clade viruses detected (Figure 1). For the libraries sequenced that reached 20× coverage for the IAV HA gene (n = 22), we compared the consensus sequences using BLAST to swine sequences submitted by the U.S. Department of Agriculture (USDA). We found that the top blast hits from the consensus sequences were highly concordant with the IAV-S triplex rRT-PCR results in the majority of cases, with the exception of sample 125, which did not give a positive Ct value on the IAV-S triplex rRT-PCR assay but was found to be an IAV-S A(H1)-1B.2.1 clade virus via sequencing (Table 6 and Appendix A). We did not observe mutations within the IAV-S triplex rRT-PCR primer and probe binding regions in the sequencing data for sample 125.

Phylogenetic analysis was performed using Molecular Evolutionary Genetics Analysis software (MEGA, version 7.0). The evolutionary history was inferred using the Maximum Likelihood method.

There were additional samples that provided a positive Ct value with the IAV-S triplex rRT-PCR but did not achieve sequencing coverage of 20× for the HA gene (n = 8; Table 6 and Appendix A). Due to the lack of sequencing coverage from these eight sequencing libraries, we were not confident in the consensus sequence derived from these samples, however, the top blast hits for the majority of these samples correspond to the same viral clades as those that the IAV-S triplex rRT-PCR assay detected. Overall, sequencing confirmed that the IAV-S triplex rRT-PCR assay was capable of providing accurate information on current influenza outbreaks occurring in exhibition swine within hours of sampling.

## 4. Discussion

The IAV-S triplex rRT-PCR assay presented here is intended to rapidly characterize circulating N.A. IAV-S using rRT-PCR technology. Many IAV-S circulating in South America, Europe, and Asia are genetically different from the IAV-S circulating in N.A., and, thus, this assay is designed specifically for N.A. IAV-S [35,36]. To distinguish HA genes of IAV-S A(H1) 1A clades from A(H1N1)pdm09 viruses, which inherited their HA gene from the IAV-S A(H1) 1A.3.3.3 (γ) clade, a thymine to cytosine mutation is introduced at the tenth nucleic acid from the 5′ end of the cH1 probe. Due to this introduced mutation, the cH1 probe does not bind A(H1N1)pdm09 and the IAV-S triplex rRT-PCR assay does not cross-react with A(H1N1)pdm09 viruses (Table 4 and Appendix A). However, this nucleic acid mutation can be corrected to increase the assay sensitivity and accommodate potential nucleotide mismatches occurring within the probe region due to virus evolution, as needed.

When IAV-S infect humans, these are currently termed variant viruses. From 2005 to August 2022, 504 influenza variant virus infections have been identified in the U.S., and the majority of these infections were due to an initial transmission of IAV-S between swine and humans at agricultural fairs [37]. These infections can be serious and lead to hospitalization or death. For example, in 2012, there were 309 human infections with A(H3N2)v, and of those infections, 16 patients were hospitalized, and one patient died [4]. In 2016, a novel A(H3N2)v, IAV-S A(H3) 2010.1 clade virus infection caused ILI in 18 patients, with one patient hospitalized due to a comorbidity [7]. The swine-human interface is a high-risk environment for viral transmissions between the two species, therefore, it is critical to monitor IAV-S that are circulating at these points. Early detection of IAV-S that has the potential to cause outbreaks and pandemics allows for the implementation of an effective public health response.

Here, we demonstrated the feasibility of utilizing the IAV-S triplex rRT-PCR assay to deliver reliable information on the IAVs present in an exhibition swine show via an in-field test. Swine is a vessel for continued generation and evolution of IAV diversity [38]. At swine exhibitions, the swine are in close contact with each other and humans, leading to a high-risk environment for viral transmissions. By performing nucleic acid extraction and using a portable real-time PCR instrument and portable sequencing methods onsite at the exhibition, within hours of taking swine nasal samples, we detected three separate IAV-S clades [A(H1) 1B.2.1, 1A.3.3.3, and 1A.3.3.2 clade viruses]. The predominant IAV-S subtype detected at the swine exhibition was IAV-S A(H1) 1A.3.3.3 clade viruses (Table 6 and Appendix A). The (H1) 1A.3.3.3 clade virus has antigenic cross-reactivity with A(H1N1)pdm09, due to A(H1N1)pdm09 inheriting the HA gene from the A(H1) 1A.3.3.3 clade virus, and therefore less of a risk of causing a human outbreak. However, we also detected a subset of IAV-S A(H1) 1B.2.1 clade viruses, which present a greater risk of causing human outbreaks due to the lack of sufficient immunity in humans to this subtype of virus, particularly in the young human population that has not been exposed to an A(H1N1) IAV circulating prior to 2009 through natural exposure or vaccination.

The rRT-PCR cycling conditions for the IAV-S triplex rRT-PCR assay are consistent with the cycling conditions for the current CDC Flu rRT-PCR Dx Panel for the detection, diagnosis, and subtyping of seasonal influenza viruses. Thus, the IAV-S triplex rRT-PCR assay can be tested together with the assays of the CDC Flu rRT-PCR Dx Panel, such as InfA, and the universal pandemic influenza A target (pdmInfA), A(H1N1)pdm09 H1 (pdmH1), and H3 assays [28,29], as well as the other CDC IVD assays to further characterize influenza virus subtypes and clades. Subsequent to this study, we updated the fluorphores and quenchers used in the IAV-S triplex rRT-PCR assay to match what was used in the Flu SC2 Multiplex assay [27], which enhanced the sensitivity of the assay (Appendix A), and we recommend this updated build for labs wanting to use this assay.

Overall, the IAV-S triplex rRT-PCR assay is a portable resource that can be easily deployed to locations of swine-human interfaces and can rapidly and sensitively monitor circulating N.A. IAV-S, including IAV-S A(H1) and A(H3) 2010.1 clades. As this is an rRT-PCR-based assay, it is more sensitive than field-deployable insulated isothermal PCR assays for detecting IAV-S in the field and provides additional information on the clade of the circulating IAV-S [39]. By coupling the IAV-S triplex rRT-PCR assay with portable methods and instrumentation, including in-field sequencing, we demonstrate that the IAV-S triplex rRT-PCR assay can efficiently and promptly detect N.A. IAV-S outbreaks in-field, ultimately allowing timely deployment of interventions to prevent virus transmission to other swine and to humans during IAV-S outbreaks.

## Figures and Tables

**Figure 1 viruses-16-01557-f001:**
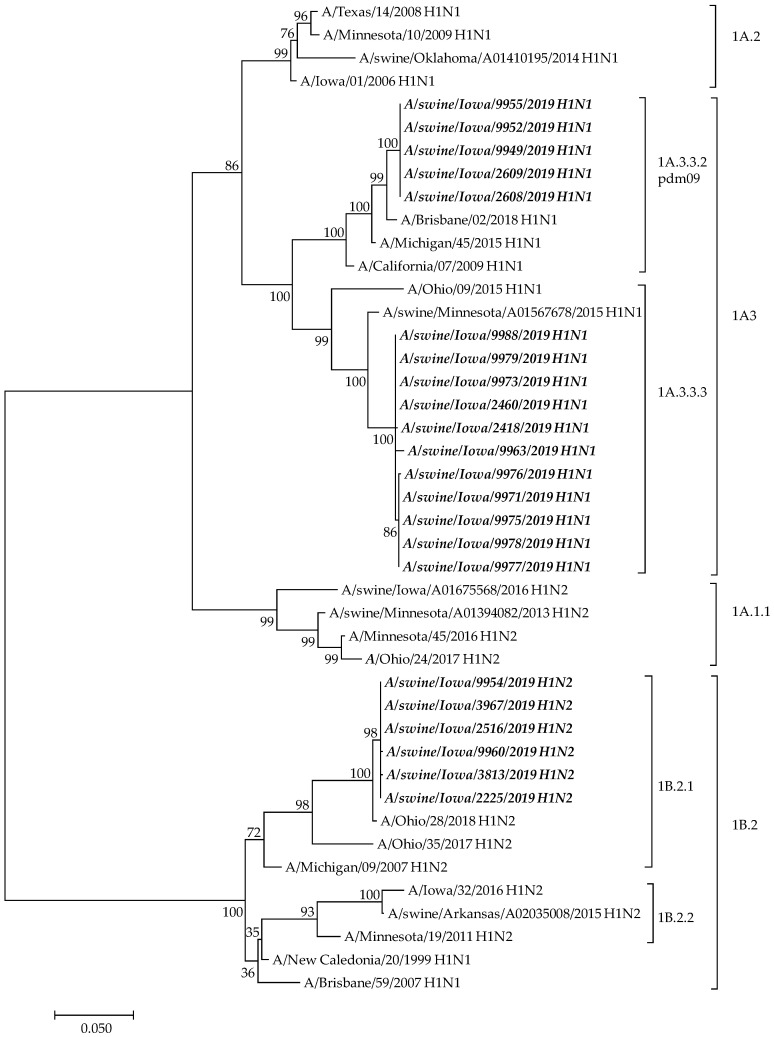
Phylogenetic analysis: H1HA genes of swine influenza samples from the exhibition swine with H1 clade of related viruses, 1A.1.1, 1A.2, 1A.3, H1 1B.2, and A(H1N1)pdm09 (1A.3.3.2) indicated by the bars on the right. The sequences obtained from this study are shown in bold and italic font. Phylogenetic analysis was performed using Molecular Evolutionary Genetics Analysis software (MEGA, version 7.0). The evolutionary history was inferred using the maximum likelihood method.

**Table 1 viruses-16-01557-t001:** Primer and probe sequences for the IAV-S triplex real-time RT-PCR assay.

Assay	Primer/Probe	Sequence 5′-3′	Nucleotide Position ^1^
sH1	Forward primer	CTT CCT TTC CAG AAY GTA CAY CC	916–938
	Reverse primer	GCT CCA AAC AAA CCT CTG GAT TG	1043–1021
	Probe	ACT CC“T” GAC TAT AYT TTG GAC AYT CTC CTA TTG ^2^	975–944
H3v_2010	Forward primer	AAG CAT TCC MAA TGA CAA ACC	906–926
	Reverse primer	ATT GCG CCR AAT ATG CCY CTA GT	1052–1030
	Probe	TGG TAT ATT TCT CAT TC ^3^	1020–1004
cH1	Forward primer_a	GTG CTA TAA ACA CCA GCC TYC CAT T	916–940
	Forward primer_b	GTG CCC TAA ACA CCA GCC TCC CAT T	916–940
	Reverse primer	CGG GAT ATT CCT TAA TCC TGT RGC	1031–1008
	Probe	TGG ACA “T”TC ***C**CC AAT TGT GAC TGG AT ^4^	977–952

^1^ Nucleotide Positions are according to the HA gene coding sequences with GISAID accession number of A/Brisbane/59/2007(H1N1) (EPI 162331), A/Ohio/28/2016(H3N2)v (EPI 824757) and A/Ohio/24/2017(H1N2)v (EPI 1056725); ^2^ The BHQ probe: FAM (single-plex, multiplex) at 5′ end and quenched with a Blackhole Quencher™ 1 internally at a modified “T” residue with the addition of a Spacer 6 (C6) modification at the 3′-end; ^3^ The BHQplus probe: FAM (single-plex) or Hex (Triplex) at 5′ end, the Blackhole Quencher™ 1 at 3′ end and employed C-5 propynyl-dC (pdC) for dC and C-5 propynyl-dU (pdU) for dT substitutions; ^4^ The BHQ probe: FAM (single-plex) and Quasar 670 (Triplex, Cy5 channel) at 5′ end and quenched with a Blackhole Quencher™ 2 internally at a modified “T” residue with the addition of a Spacer 3 (C3) modification at the 3′ end; ***C** was introduced from a thymine to cytosine at the tenth nucleic acid from the 5′ end of the probe.

**Table 2 viruses-16-01557-t002:** Assay limit of detection with swine influenza H1 and H3_2010.1 clade viruses (N = 3).

IAV-S Triplex rRT-PCR Component	Swine Influenza A Virus	Subtype/Clade	Limit of Detection (EID_50_/_mL_)	Ct Value (Mean ± SD)	
				InfA	IAV-S Triplex
sH1	A/Ohio/35/2017	H1N2v/1B.2.1	10^3.9^	30.79 ± 0.30	31.22 ± 0.25
H3v_2010	A/Ohio/27/2016	H3N2v/2010.1	10^4.1^	22.64 ± 0.04	28.74 ± 0.06
cH1	A/Ohio/24/2017	H1N2v/1A.1.1	10^3.3^	28.06 ± 0.44	30.41 ± 1.63

**Table 3 viruses-16-01557-t003:** Analytical specificity testing with human A(H1N1), swine influenza A(H1), and A(H3) viruses isolated from swine or human samples (N = 3).

Influenza A Viruses	Subtype/Clade	Infectious Titer (ID_50_/mL)	Ct Value of rRT-PCR (Mean ± SD)			
			InfA	IAV-S Triplex rRT-PCR		
				cH1	sH1	H3v 2010
A/Brisbane/59/2007	Human H1N1/1B.2	10^5.4 a^	21.25 ± 0.20	-	23.71 ± 0.31	-
A/Maryland 12/1991	H1N1v/1A.1	10^8.2 a^	18.76 ± 0.25	20.22 ± 0.62	-	-
A/Minnesota/45/2016	H1N1v/1A.1.1	10^7.9 a^	17.41 ± 0.11	19.14 ± 0.18	-	-
A/Ohio/24/2017	H1N2v/1A.1.1	10^6.3 b^	18.38 ± 0.26	19.09 ± 0.03	-	-
A/Thailand/271/2005	H1N1v/1A.1.4	not available	16.82 ± 0.29	-	-	-
A/Texas/14/2008	H1N1v/1A.2	10^8.3 a^	15.65 ± 0.12	18.90 ± 0.10	-	-
A/Iowa/1/2006	H1N1v/1A.2	10^8.2 a^	16.31 ± 0.05	19.50 ± 0.96	-	-
A/Ohio/09/2015	H1N1v/1A.3.3.3	10^7.7 b^	14.91 ± 0.10	17.27 ± 0.05	-	-
A/Iowa/32/2016	H1N2v/1B.2.2	not available	16.34 ± 1.05	-	16.29 ± 0.45	-
A/Minnesota/19/2011	H1N2v/1B.2.2	10^7.1 b^	17.74 ± 0.31	-	18.96 ± 0.41	-
A/Ohio/35/2017	H1N2v/1B.2.1	10^6.9 b^	16.95 ± 0.22	-	17.52 ± 0.38	-
A/Minnesota/11/2010	H3N2v/1990.4	10^8.2 a^	17.55 ± 0.28	-	-	-
A/West Virginia/06/2011	H3N2v/1990.4	10^5.9 b^	17.79 ± 0.93	-	-	-
A/Ohio/13/2017	H3N2v/2010.1	10^6.6 b^	16.81 ± 0.39	-	-	19.47 ± 1.23
A/Ohio/28/2016	H3N2v/2010.1	10^9.2 a^	18.34 ± 1.46	-	-	20.85 ± 1.30
A/swine/Oklahoma/A02218157/2017	H3N2v/2010.2	not available	19.60 ± 0.41	-	-	-

^a^ EID_50_/mL; ^b^ TCID_50_/mL.

**Table 4 viruses-16-01557-t004:** Analytical specificity testing with seasonal H3N2, H1N1pdm09, highly pathogenic avian influenza (HPAI) H5N1, Eurasian lineage H7N9 influenza A and B viruses (N = 3).

Influenza Virus	Subtype or Lineage	Infectious Titer (EID_50_/mL)	Ct Value of rRT-PCR (Mean ± SD)		
			InfA	InfB	IAV-S Triplex
A/Michigan/45/2015	H1N1pdm09	8.3	16.17 ± 0.20	-	-
A/Illinois/20/2018	H1N1pdm09	6.8	20.78 ± 0.47	-	-
A/Hong Kong/4801/2014	H3N2	8.4	12.64 ± 0.29	-	-
A/Abu Dhabi/240/2018	H3N2	8.1	16.08 ± 0.21	-	-
A/duck/Vietnam/NCVD-1544/2012	H5N1	8.5	16.01 ± 0.16	-	-
A/Anhui/1/2013	H7N9	10.9	13.54 ± 0.03	-	-
A/Taiwan/1/2017	H7N9	8.5	14.69 ± 0.32	-	-
B/Maryland/15/2016	B/Victoria	6.5	-	17.08 ± 0.13	-
B/Texas/81/2016	B/Yamagata	6.3	-	17.34 ± 0.52	-

**Table 5 viruses-16-01557-t005:** IAV-S triplex rRT-PCR assay comparison with AB 7500 Fast Dx real-time PCR and the Quantabio “Q” qPCR instrument (N = 2).

Influenza Virus	Subtype/Clade	Infectious Titers (EID^50^/mL)	sH1 (FAM)		H3v_2010(Hex)		cH1 (Cy5)	
			AB 7500	“Q” qPCR	AB 7500	“Q” qPCR	AB 7500	“Q” qPCR
A/Ohio/35/2017	H1N2v/1B.2.1	10^5.9^	20.40 ± 0.01	20.86 ± 0.01	-	-	-	-
A/Brisbane/59/2007	human H1N1	not available	23.64 ± 0.09	24.06 ± 0.03	-	-	-	-
A/Ohio/27/2016	H3N2v/2010.1	10^5.1^	-	-	24.72 ± 0.16	23.58 ± 0.09	-	-
A/Ohio/28/2016	H3N2v/2010.1	not available	-	-	29.78 ± 0.02	28.08 ± 0.09	-	-
A/Ohio/24/2017	H1N2v/1A.1.1	10^5.3^	-	-	-	-	21.79 ± 0.02	22.60 ± 0.14
A/Texas/14/2018	H1N1v/1A.2	not available	-	-	-	-	26.93 ± 0.00	26.66 ± 0.07

**Table 6 viruses-16-01557-t006:** Field performance of the IAV-S triplex rRT-PCR from exhibition swine sampling vs. in-field sequencing (N = 60).

		Swine Influenza Virus Clade ^2^					Total
		H1 1B.2.1	H1 1A.3.3.3	H3 2010.1	(H1N1) pdm09	No Sequence	
**IAV-S-Triplex rRT-PCR**	sH1	5	0	0	0	1	6
	cH1	0	11	0	0	7	18
	H3v 2010	0	0	0	0	0	0
	Negative ^1^	1	0	0	5	30	36
Total		6	11	0	5	38	60

^1^ All three targets of IAV-S-triplex rRT-PCR, including sH1, cH1, and H3v 2010, tested negative; ^2^ Swine influenza virus clades were determined from HA top blast hits for the consensus sequence at 20× average coverage.

## Data Availability

Related sequencing data is available here: Influenza A virus (ID 1153990)—BioProject—NCBI (nih.gov; accessed on 10 September 2024).

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
