# Peer review of "Discriminating North American Swine Influenza Viruses with a Portable, One-Step, Triplex Real-Time RT-PCR Assay, and Portable Sequencing"

_viruses, 2024, doi:10.3390/v16101557_

Round 1

Reviewer 1 Report

Comments and Suggestions for Authors

Reviewer comments:

The manuscripts reports on developing assays and potable sequencing for swine influenza viruses that are mainly targeting for North America. The data are of interest for the influenza virus research community and the readers of Virology. The study has been well conducted and the manuscript is well written. I have only a few major comments and suggestions, mainly to improve the clarity and representation of the data for the reader, make them more digestible and easier to interpret.

Major comments:

1)        The Authors should describe how they will utilize the assay. The assay which the authors developed is not effective to detect some clades like 1A.3.3.2, H3N2v/1990.4 and H3N2v/2010.2. If H3N2/1990.4 and H3N2.2010.2 IAVs-S are rarely isolated lately and can be ignored for some reason, the authors should describe the current situation of these clades. On the other hand, 1A.3.3.2 IAVs-S are now circulating globally and actually isolated in the field study as described in Table S3. Thus, the description regarding how the author utilize the method to detect/ignore the clades and/or complement by other methods.

Minor comments and suggestions:

A)      Throughout the text: I have seldomly seen “IAV-Ss” but “IAVs-S” or “IAV-S strains”. IAV-Ss needs to be revised.

B)      Lines 37-38: Maybe double space is existing between “viruses” and “from”.

C)      Lines 115-124: In the study using the sequences of the database, information regarding the conditions for source is important. The authors should write details of downloading like “When the authors download”, “How many sequences the authors downloaded”, “How the authors filtered the downloaded sequences to use them”, and “How many sequences the author used eventually to arrange the primer/probes” for each set.

D)      Table S2: The font of 10-3 and corresponding Ct value in A/Indiana/27/2018(H3N2)v 2010.1 should be fixed.

E)       Line 306:  No space between “A/Ohaio/27/2016” and “and”.

F)       Lines 347: Maybe double space is existing between “sequences” and “using”.

G)      Figure 1: The phylogenetic tree for influenza viruses should be constructed by the Maximum likelihood method than the Neighbor-Joining method.

H)      Figure S2 and Table S3: These are confusing. Why the authors tested 3 samples not tested in night 1 (but the authors tested 92/96 samples in night 1, remaining 4 samples)? The authors should describe the difference in sensitivity between Single-Triplex test as at least 21 positive samples in night 1 (Singleplex) was 14 positives in night 2 (Triplex RT-PCR). Which samples in Table S3 are of the night 1 (and tested positive in Singleplex)?

Comments on the Quality of English Language

The minor corrections are needed as described in the comments and suggestions for Authors.

Author Response

Reviewer 1:

Thank you so much for your review and comments on our manuscript. Please find detailed responses below and the corresponding revisions/corrections highlighted in track changes in the re-submitted files.

Major comments:

1)        The Authors should describe how they will utilize the assay. The assay which the authors developed is not effective to detect some clades like 1A.3.3.2, H3N2v/1990.4 and H3N2v/2010.2. If H3N2/1990.4 and H3N2.2010.2 IAVs-S are rarely isolated lately and can be ignored for some reason, the authors should describe the current situation of these clades. On the other hand, 1A.3.3.2 IAVs-S are now circulating globally and actually isolated in the field study as described in Table S3. Thus, the description regarding how the author utilize the method to detect/ignore the clades and/or complement by other methods.

Thank you for this comment. We developed this assay based on the major clades found to be circulating in north American swine. To add clarity to our manuscript, we have included an additional reference into the paper that describes influenza with pandemic potential. Reference 25: [25] WHO, Antigenic and genetic characteristics of zoonotic influenza viruses and development of candidate vaccine viruses for pandemic preparedness. https://www.who.int/docs/default-source/influenza/who-influenza-recommendations/vcm-southern-hemisphere-recommendation-2019/201809-zoonotic-vaccinevirusupdate.pdf. 2018. Accessed 9/25/2024. We have developed additional assays to detect 1A.3.3.2, H3N2v/1990.4 and H3N2v/2010.2, but those assays were not the focus of this manuscript.

Minor Comments and suggestions:

  1. Throughout the text: I have seldomly seen “IAV-Ss” but “IAVs-S” or “IAV-S strains”. IAV-Ss needs to be revised.

Thank you – we have updated the text to remove IAV-Ss – we replaced with IAV-S.

  1. Lines 37-38: Maybe double space is existing between “viruses” and “from”.

Thank you for this observation. We have removed the space between “viruses” and “from” (lines 37-38 in the original submission).

  1. Lines 115-124: In the study using the sequences of the database, information regarding the conditions for source is important. The authors should write details of downloading like “When the authors download”, “How many sequences the authors downloaded”, “How the authors filtered the downloaded sequences to use them”, and “How many sequences the author used eventually to arrange the primer/probes” for each set.

To clarify for readers, we added “each available clade”…”at the time of development of this assay” on line 129-132. We also included an additional statement on lines 131-133 to describe the process more clearly: All available HA sequences for each clade were downloaded and aligned. Poor quality sequences were removed.

  1. Table S2: The font of 10-3and corresponding Ct value in A/Indiana/27/2018(H3N2)v 2010.1 should be fixed.

Thank you for pointing this out. 10-3 and Ct value 27.95 in Table S2 have been fixed.

  1. Line 306:  No space between “A/Ohaio/27/2016” and “and”.

We have added a space between “A/Ohio/27/2016” and “and”.

  1. Lines 347: Maybe double space is existing between “sequences” and “using”.

We have removed the space between “sequences” and “using”.

  1. Figure 1: The phylogenetic tree for influenza viruses should be constructed by the Maximum likelihood method than the Neighbor-Joining method.

We have reconstructed phylogenetic tree by the Maximum Likelihood method and updated phylogenetic analysis method from “Neighbor-Joining method” to “Maximum Likelihood method” in figure 1 legend.

  1. H)Figure S2 and Table S3: These are confusing. Why the authors tested 3 samples not tested in night 1 (but the authors tested 92/96 samples in night 1, remaining 4 samples)? The authors should describe the difference in sensitivity between Single-Triplex test as at least 21 positive samples in night 1 (Singleplex) was 14 positives in night 2 (Triplex RT-PCR). Which samples in Table S3 are of the night 1 (and tested positive in Singleplex)?

Since we performed the test on 2 instruments [one instrument has 48 wells, we tested 46 samples in each instrument since we had to include one No Template Control (NTC) and Influenza A Positive Control (InfA PC), thus we could test 92 samples only on day 1.  We tested the additional 3 samples on day 2 but we did not include the 4th sample due to a known user error with that specimen during extraction. To clarify that, we have added a sentence into section 3.3: The fourth sample from day 1 was not included in day 2 testing due to a known user error. 

To clarify the reviewer comment about 21 positives for InfA on day 1 but 14 specimens positive for the swine triplex on day 2 –  the day 1 samples are the first 21 samples in the table S3. Two were shown to be (H1N1)pdm09 through sequencing, and the swine triplex doesn’t detect that virus purposefully. The other 5 specimens that were positive for InfA but negative in swine triplex had very weak Ct value for InfA (mid to late 30s) so these were unable to be subtyped or sequenced successfully.

Reviewer 2 Report

Comments and Suggestions for Authors

Major comments

Abstract

-        You should report a summary of the materials and methods of your study

Introduction

-        L95-104: You should more clearly refer to the aim of this article. You should not report the results of your study in the section of ‘’introduction’’

Materials and Methods

-        L217-221: provide the approval number for this study by an ethical committee from your institution or university 

Discussion  

-        You could add a paragraph, underlying the importance of your results for future control strategies against Swine Influenza Viruses.

Minor comments

·       You should improve issues of plagiarism (Percent match: 35%)

·       L58: In the mid-2000s,

·       L67: .. Currently,  at least 10 distinct HA genetic clades are co-circulating in N.A.

·       L80: circulated before 2009

·       L282: .. cross-reactivity..

·       L322: .. in a viral transport medium

·       L337: .. and 6 swine were positive

Author Response

Reviewer 2:

Thank you so much for your review and comments on our manuscript. Please find detailed responses below and the corresponding revisions/corrections highlighted in track changes in the re-submitted files.

Major comments:

Abstract

-        You should report a summary of the materials and methods of your study

Thank you – we have updated our abstract to hopefully more clearly define the materials and methods. Abstract now reads:

Swine harbor a genetically diverse population of swine influenza A viruses (IAV-S), with demonstrated potential to transmit to the human population, causing outbreaks and pandemics. Here, we describe development of a one-step, triplex real-time reverse transcription-Polymerase Chain Reaction (rRT-PCR) assay that detects and distinguishes the majority of the antigenically distinct influenza A virus hemagglutinin (HA) clades currently circulating in North American swine, including the IAV-S H1 1A.1(α), 1A.2 (β), 1A.3 (γ), 1B.2.2 (δ1) and 1B.2.1 (δ2) clades, and the IAV-S H3 2010.1 clade. We perform an in-field test at an exhibition swine show using in-field viral concentration and RNA extraction methodologies and a portable Real-Time PCR instrument, and rapidly identified three distinct IAV-S clades circulating within the N.A. swine population. Portable sequencing is used to further confirm the results of the in-field test of the swine triplex assay. The IAV-S triplex rRT-PCR assay can be easily transported and used in-field to characterize circulating IAV-S clades in North America, allowing for surveillance and early detection of North American IAV-S with human outbreak and pandemic potential.

Introduction

-        L95-104: You should more clearly refer to the aim of this article. You should not report the results of your study in the section of ‘’introduction’’

Thank you – we removed the following verbiage from the introduction section: We adapted the instrumentation to mobilize this assay for in-field use directly at interfaces of high zoonotic transmission potential, such as agricultural fairs and swine exhibitions. We demonstrate the portability of this assay via an in-field test at an exhibition swine show. We used in-field viral concentration and RNA extraction methodologies and a portable Real-Time PCR instrument, the Quantabio “Q” qPCR instrument, and rapidly identified three distinct IAV-S clades circulating within the N.A. swine population.

Materials and Methods

-        L217-221: provide the approval number for this study by an ethical committee from your institution or university

Thank you – we have included the following verbiage to a section at the end of the manuscript (line 534-535): Project determination number: virology, surveillance, and diagnosis branch project determination number 0900f3eb8231bdeb

Discussion  

-        You could add a paragraph, underlying the importance of your results for future control strategies against Swine Influenza Viruses.

Thank you for this comment. While an important topic for discussion, we feel that control strategies against swine influenza viruses is a topic that is outside the scope of this manuscript. We opted to use the discussion section to discuss how this portable RT-PCR assay can be used at points of human-swine interface for rapid detection of swine influenza outbreaks.

Minor comments

  • You should improve issues of plagiarism (Percent match: 35%)

We appreciate your vigilance to plagiarism. It is difficult to respond as it is unclear what tool/algorithm was used in this evaluation. We certify that this is original work and the manuscript was drafted by the authors with no plagiarism.

  • L58: In the mid-2000s,

We have updated “2000’s” to “2000s”

  • L67: .. Currently,  at least 10 distinct HA genetic clades are co-circulating in N.A.

We have removed “there are” between “currently” and “at least…”

  • L80: circulated before 2009

We have updated “prior to“ to “before”

  • L282: .. cross-reactivity..

We have updated “cross reactivity” to “cross-reactivity”.

  • L322: .. in a viral transport medium

We have added “a” into “in viral transport medium”.

  • L337: .. and 6 swine were positive

We have removed” that” between “6 swine” and “were positive”.